# Lightweight Multimechanism Deep Feature Enhancement Network for Infrared Small-Target Detection

**Yi Zhang** [1,†] , **Bingkun Nian** [2,†] , **Yan Zhang** [1,*], **Yu Zhang** [1] and **Feng Ling** [1]

1 National Key Laboratory of Science and Technology on Automatic Target Recognition, Collage of Electronic Science and Technology, National University of Defense Technology, Changsha 410073, China
2 The 32nd Institute of China Electricity Science and Technology, Shanghai 210808, China
* Correspondence: atrthreefire@nudt.edu.cn
† These authors contributed equally to this work.

**Abstract:** Specific to the problem of infrared small-target detection in complex backgrounds, a multimechanism deep feature enhancement network model (MDFENet) was proposed. A lightweight multimechanism attention collaborative fusion module was proposed to efficiently fuse low-level features and high-level features to solve the problem that small infrared targets are easy to annihilate in the deep layer of the network. Based on the analysis of the background and target data, a normalized loss function was proposed, which integrates the segmentation threshold selection into the network and normalizes the probability of the network output to simulate a step function and reflect relative differences. Aiming at the sparseness of infrared target features, we used the subpixel convolution method to upsample the features to obtain high-resolution feature images while expanding the size of the feature map. We conducted detailed comparison and ablation experiments, comparing MDFENet with ALCNet, APGCNet, and other state-of-the-art networks to verify the effectiveness and efficiency of the network. Results show that the MDFENet algorithm achieves the optimal result in the balance of detection efficiency and lightweightedness on two datasets.

**Keywords:** small infrared target detection; lightweight; multimechanism attention collaborative fusion module; normalized loss function

## 1. Introduction

Infrared small-target detection is a significant challenge faced in infrared image processing. It is widely used in many fields such as precision guidance [1], early warning systems [2], and maritime surveillance [3]. Compared with other imaging methods, infrared imaging has the characteristics of long detection distance, all-weather application, strong anti-interference ability, and clear images [4]. However, small infrared targets often have fewer pixels, and the characteristics of shape, texture, and color are missing. Moreover, the target is often submerged in complex backgrounds and greatly influenced by changes in the surrounding environment. Therefore, separating infrared small targets from complex background clutter is challenging and has attracted extensive research in recent years.

There are two main methods of infrared small-target detection: model-driven and data-driven methods. Model-driven methods were developed in depth in the past decades, and several types of systematic methods have been proposed. The main methods are based on background suppression [5–8], human visual system [9–13], and optimization [14–16]. Background suppression-based methods consider the target to break the continuity of the image [17], and use filtering and morphological methods to suppress the background and separate the targets. Human visual system methods assume that there is a large local contrast between the target and the background; therefore, areas with high local contrast may be targets. Optimization-based methods adopt the concept of the matrix. That is, the target analogy is a sparse matrix, and the background analogy is a low-rank matrix, continuously optimizing the separation of low-rank and sparse matrices to achieve target

detection. However, these methods focus on the intrinsic characteristics of the target, adopt different assumptions, rely heavily on prior knowledge, and manually set functions. They have difficulties in dealing with changes in the characteristics of real scenes. When a real scene does not reach the detection conditions, the detection results will be poor. Despite the continued development of new algorithms, there are still problems such as low detection accuracy, high environmental impact, and poor robustness [18].

In contrast with model-driven methods, data-driven methods have not developed rapidly due to the lack of datasets but have achieved better results than model-driven methods. In infrared small-target detection networks, researchers focus on introducing attention mechanisms [17], combining model- and data-driven methods [1,19] and multiscale feature fusion [20]. Other methods—asymmetric context modulation module (ACM), cascaded channel and spatial attention module (CSAM), attention guided context block (AGCB), and feature pyramid network(FPN)—have immensely contributed to the development and prosperity of infrared small-target detection. Although data-driven infrared small-target detection has been developed from different angles, most of the methods still use general target detection algorithms without improving the loss and other modules of the adjusted algorithm, ignoring the inherent physical characteristics of infrared small targets. Moreover, small infrared targets are very different from general targets in terms of size and color characteristics. Directly applying these methods without combining the physical characteristics of small infrared targets for infrared small-target detection tends to lead to the loss of deep small targets.

To overcome the above defects, inspired by [19], we proposed a lightweight multi-mechanism deep feature enhancement network (MDFENet) algorithm. The motivations of our method, based on data analysis, can be summarized as follows.

1. The target occupies a small proportion of the whole infrared image (generally less than 0.12%) and lacks color and fine structure information (e.g., contour, shape, and texture).Therefore, it is not advisable to blindly increase the network depth and increase the extraction ability of small target features. This will not only increase the computational burden, but also make the model file increase rapidly.

2. Using traditional loss functions, such as Soft-IoU, and using marked images (the background is 0, and the target is 1), essentially defines a fixed threshold of 0.5. However, because of the lack of infrared small target features, complex background, and variety of small targets, the network may not have the ability to make the target approach 1 and the background approach 0. For example, the background may be 0.2, and the target, 0.5. Thus, we must relativize the difference between the background and the target and simulate the effect of the step function.

3. Unlike the segmentation of large targets, in the segmentation of small infrared targets, the target features may be submerged or may disappear in the clutter in the deep layer of the network. Moreover, we believe that shallow features may play a more important task in the recognition of small infrared targets. Therefore, we must detect targets in low-level feature maps and enhance high-level semantic features.

4. Infrared small target segmentation is primarily used in the military field, which requires a small number of network calculations and a small model file. Therefore, we must design a lightweight network.

Based on the above motivation, the main contributions of our proposed method are as follows.

1. In this paper, ResNet20 is selected as the backbone network, and the encoder–decoder structure is adopted. There are shortcut connections used in ResNet20. These shortcut connections can fuse features with different resolutions to a certain extent, which helps to alleviate the deep annihilation problem of infrared small target and keep the gradient stable. To minimize unnecessary floating-point operations, we refer to ALCNet and reduce the four-layer structure of the traditional ResNet to three layers. Thus, the floating-point operation is controlled at 4.3 G, and the model size is 0.37 M. It can be embedded in aviation equipment.

2.  A multimechanism attention collaborative fusion module is proposed, including a weak target channel attention mechanism and pixel attention mechanism. Most of the existing modules are devoted to the development of more complex attention mechanisms, which are not suitable for infrared small-target detection. Lightweight attention mechanisms, such as efficient channel attention network (ECA-Net) [21] cannot effectively form attention to small targets because of the lack of some features of infrared small targets. To solve the above problems, the multimechanism attention collaborative fusion module adopts the concept of multimechanism collaborative feature fusion under the guidance of an attention mechanism and makes improvements in target detection in two aspects. First, a global maximum pooling and sigmoid function are used to enhance the semantic features of small targets and an uncompressed channel convolution is used to extract the channel attention, which effectively retains deep semantic information. Secondly, under the guidance of deep semantic features, the detail features of infrared small targets are retained and enhanced by pixel attention, and the underlying features are modulated by multiplying elements so that the network dynamically selects relevant features from the bottom.

3.  The loss function in the general network is no longer suitable for the end-to-end detection method that combines model- and data-driven methods. Therefore, we propose a normalized loss function. It normalizes the network output with the maximum and minimum probability, ensures that the loss function focuses on the relative value between pixels, and improves the convergence efficiency of the network, which is also a key factor in improving the efficiency of the model- and data-driven end-to-end detection methods.

4.  To address the problem that the small infrared target has few pixels and the difficulty in extracting high-resolution effective features, subpixel convolution upsampling is used to enhance the feature utilization and the low-resolution feature representation is converted into high-resolution identifiable feature representation to improve the detection accuracy of the network.

To evaluate the effectiveness of MDFENet, we conducted several ablation experiments on different network layers. We also compared MDFENet with several model-driven and advanced data-driven methods on the open SIRST dataset [22] and the improved IDTAT dataset [23]. Furthermore, we made improvements to the IDTAT dataset: from the original 22 real infrared sequences, 10 infrared sequences with complex backgrounds and weak targets were selected and the targets were marked at pixel level. The 10 sequence images were combined to form an experimental dataset. The experimental results showed that each module of MDFENet is effective and that the normalized loss (NL) function can more accurately adapt to the combination of model- and data-driven methods.

## 2. Related Work

### 2.1. Single-Frame Infrared Small-Target Detection

As a difficult task in target detection, small-target detection has been of major concern to researchers. With many research achievements, infrared small-target detection is an important branch of small-target detection. The traditional single-frame infrared small-target detection focuses on locating the target position and regards the detection problem as an image-anomaly detection problem under various assumptions. Typical methods include a detection algorithm based on filtering [24,25], local contrast [9,10], and image data structure [14,15]. These methods are generally based on the gray feature difference between the small target and background. A saliency map can be obtained by saliency detection, local contrast measurement, sparsity, and low-rank matrix factorization. Adaptive threshold segmentation is then used to separate infrared small targets from saliency images. LCM [9] achieves target enhancement and background suppression by redefining contrast by the ratio of the maximum value of the central subblock to the mean value of the eight directional subblocks in the field. MPCM [10], on this basis, defines the contrast difference to achieve the enhancement of light and dark targets. FKRW [26] uses the

heterogeneity and compactness of the target region to achieve target enhancement and clustering suppression and combines the random walk algorithm and facet kernel filter to construct a new local contrast operator for small-target detection. IPI [14] combines local patch construction with the optimization of separated low-rank matrix and sparse matrix to obtain stronger robustness. PSTNN [18] combines the rank of the partial sum of tensor nuclear norm as a low-rank constraint and effectively maintains the goal. Although the traditional single-frame infrared small-target detection has been greatly developed and has the characteristics of a simple algorithm and easy implementation, the features designed according to prior knowledge are not as robust to changes in diversity; it is difficult to adapt to the complex background and too sensitive for the hyperparameters of the algorithm. To address this problem, researchers applied deep learning methods that have made breakthrough progress in image classification and target detection to infrared small-target detection and their performance was superior to traditional methods. The first infrared small-target detection algorithm based on a neural network was proposed by Liu et al. [27]. They detected targets by using a five-layer multilayer perception network. The first segmentation-based infrared small-target detection method was proposed by Dai et al. [22]. In contrast with traditional detection methods, which detect whether there are targets at a specific location, segmentation-based infrared small-target detection can accurately segment small targets through pixel-level annotation on the dataset to achieve higher detection accuracy. They designed an asymmetric context module (ACM) to achieve a cross-layer fusion, which uses the asymmetric structure to fuse the underlying semantics with the deep semantics to obtain more efficient feature representation. On this basis, Dai et al. further proposed the ALCNet algorithm [19], which combines the advantages of model-driven and data-driven models to overcome the problem of infrared small targets lacking fixed features. By using generative adversarial networks, Wang et al. [28] divided the generator into two subtasks of missed detection and false alarms and proposed MDvsFA cGan to detect small targets. Zhao et al. [29] proposed IRSTD-GAN, which takes infrared small targets as special noise and predicts them from the input image based on the data distribution and hierarchical features learned by generative adversarial networks.

### 2.2. Multiscale Feature Fusion Guided by Attention Mechanism

In neural networks, different layers retain different image features: the bottom layer retains detailed features, such as edges and contours, which are conducive to target positioning, and the deep layer retains abstract semantic features, which are conducive to the understanding of the target. To achieve more accurate positioning and segmentation, advanced networks adopt multiscale feature-fusion strategies, which fuse low-level detailed features with deep semantic features to form feature representations with fine-grained information and rich semantic information. Ronneberger et al. [30] proposed the U-NET network, which extracted features through an encoder–decoder structure and fused low- and high-level features through channel splicing. Lin et al. [31] proposed FPN, upsampling the deep features followed by top-down connections to the underlying features to achieve element-level addition, which makes full use of the bottom positioning details based on obtaining the deep semantic information and greatly improves the ability of small-target detection. Moreover, because of the excellent performance of the attention mechanism, Li et al. [32] introduced the attention mechanism based on FPN and weighted the channels of the underlying feature graph after the deep features were globally pooled to guide the fusion of information at different levels.

Because of its excellent performance, the attention mechanism is widely used in neural networks, and various attention mechanisms continue to be developed, such as SEnet [33], CBAM [34], ECA-Net [21], and SCNet [35]. SEnet generates the optimal feature map by squeeze and excitation. CBAM enhances feature maps from two perspectives, through a channel attention module and a spatial attention module. ECA-Net uses the 1D convolution generating channel attention mechanism to reduce the computation while maintaining

a good enhancement effect. SCNet uses a self-correcting convolution to realize feature enhancement without additional parameter numbers.

*2.3. Loss Function*

The loss function is an indispensable part of neural networks, which is used to measure the gap between the output value of the network and the real value and show the direction of model optimization. Most neural networks are typically trained by using a simple loss function (such as Softmax loss) or a loss function that directly optimizes a specific performance metric (such as mAP optimization). In infrared small-target detection tasks, the part of the image pixels occupied by the target is very small, and the pixel ratio between the target and the background is seriously unbalanced. Therefore, the method of directly optimizing the specific performance index is better than the method of optimizing the simple loss function. This study uses a direct optimization method for the specific performance index intersection over union (IoU).

As for the direct optimization method of the IoU index, Yu et al. [36] proposed an IoU loss function for object-detection tasks. Taking the IoU index as the loss function, the predicted value and the real value are approximated continuously. Rezatofighi et al. [37] proposed the generalized IoU loss function to solve the problem of the predicted value of the IoU loss function not reflecting the distance between the real value and the predicted value when these did not intersect. The generalized IoU loss function not only focuses on the overlap region between the network output value and the real value, but also on other noncoincidental regions to better reflect the degree of overlap between the two. Zheng et al. [38] proposed the distance-IoU loss function, which further considered the distance between the predicted value and the true value, and the overlap rate and the scale, to make the predicted value regression more stable. They further proposed the complete-IoU loss function based on the distance-IoU loss function, which considered the aspect ratio to improve the convergence speed and improve the model performance. In the segmentation task, Rahman et al. [39] proposed the soft-IoU loss function, a direct IoU optimization method, which measures the coincidence degree between the network output value and the real value through the pixel-level IoU score, and achieved good performance in the segmentation task.

## 3. Proposed Method

The current infrared small target segmentation faces a very difficult problem with regard to how to improve accuracy when performing network migration. Researchers continue to increase the number of layers of the network and add attention modules to improve the performance of the network. However, the performance improvement from these methods is minimal because of the few characteristics of infrared small targets, extremely complex background environment, and many types of small targets. Consequently, the amount of calculation and the size of the network increase rapidly. We believe that this is because of weak target learning, which tends to lead to the fixed distribution of the network. Thus, to solve this problem, we pay more attention to the overall image features in the network design, not just the individual infrared small target features.

*3.1. Network Structure*

The overall network structure of MDFENet is shown in Figure 1. The network is primarily composed of an improved ResNet-20 backbone network, multiscale local contrast (MLC) module, subpixel convolution upsampling, MAFM, and FPN mechanism. First, in the coding phase, the infrared image I was input into the improved ResNet-20 backbone network. The detailed configuration of the backbone network is shown in Table 1. The feature map is obtained through two downsamplings in the first layer of Stages 2 and 3. Thereafter, in the phase of nonlinear feature extraction, the feature map is input into the MLC module, which is proposed by ALCNet. Through slicing and cyclic displacement, the feature maps x1, x2, and x3 with fused local contrast are obtained. In the decoding

phase, the feature map is upsampled through subpixel convolution, and the deep feature map is upsampled to the same size as the feature map of the previous layer, and then input into the MAFM module. MAFM is introduced in detail in Section 3.2, which includes the weak target channel attention mechanism and the pixel attention mechanism. The two attention mechanisms enhance deep semantic features and low-level detailed features, respectively. The upsampled deep feature map has the same size as the feature map of the previous layer. After the weak target channel attention mechanism, it can be fused with the previous layer feature map enhanced by the pixel attention mechanism by using the element-by-element multiplication method. After two upsamples and the MAFM module, the predicted feature map is obtained, and the size of the predicted feature map is equal to the original image. Finally, the binary map is output in the predicted feature map to realize infrared small-target detection.

All the above modules consider the overall characteristics, not just the characteristics of a single infrared small target. To some extent, this alleviates the difficulty of infrared small targets with few features and complex environments. In addition, to avoid small-target features being submerged by the background due to convolution in the deep network, we reduce the number of subsampling layers as much as possible.

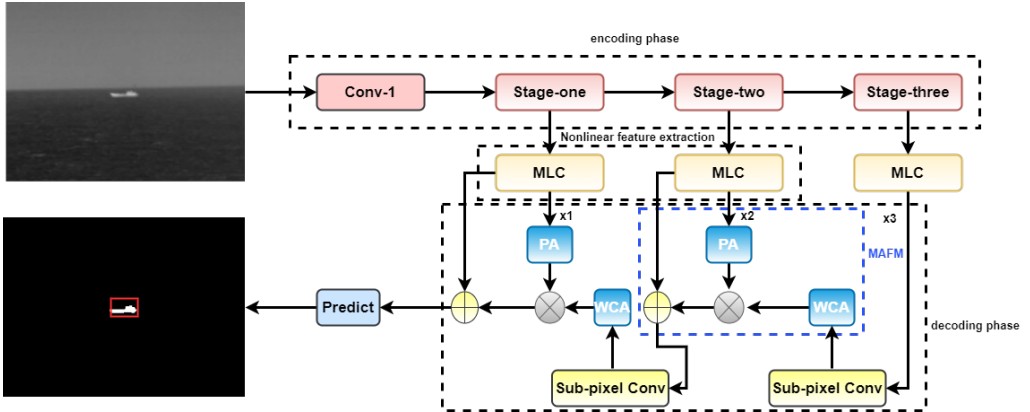

**Figure 1.** Overall structure of the MDFENet network.

**Table 1.** MDFENet backbones.

| Stage | Output | Backbone |
|---|---|---|
| conv-1 | $256 \times 256$ | $3 \times 3\mathrm{conv}, 16$ |
| Stage-one | $256 \times 256$ | $\begin{bmatrix} 3 \times 3\mathrm{conv}, 16 \\ 3 \times 3\mathrm{conv}, 16 \end{bmatrix} \times b$ |
| Stage-tow | $128 \times 128$ | $\begin{bmatrix} 3 \times 3\mathrm{conv}, 32 \\ 3 \times 3\mathrm{conv}, 32 \end{bmatrix} \times b$ |
| Stage-three | $64 \times 64$ | $\begin{bmatrix} 3 \times 3\mathrm{conv}, 64 \\ 3 \times 3\mathrm{conv}, 64 \end{bmatrix} \times b$ |

### 3.2. Multimechanism Attention Collaborative Fusion Module (MAFM)

Inspired by EAC [21] and FFA-NET [40], we propose a new attention-guided feature fusion module MAFM to fuse the underlying details and deep semantic information. The module consists of two parts: WCA and PA, and the feature fusion is performed by elementwise addition.

Although our MAFM fusion module looks very similar to BLAM (used in ALCNet [19]) and ACM [22], it is fundamentally different in terms of the design concept. Our proposed MAFM tends to use the form of feature enhancement, whereas ACM is an equal fusion of low-level semantics and high-level semantics. This is the essential difference. Moreover, the difference between MAFM and BLAM lies in the understanding of low-level semantics

and high-level semantics. We believe that low-level information shows more location information of small targets, whereas high-level information represents the possibility of existence. Therefore, it is very reasonable to use low-level semantics as the base. Moreover, because of the sparse infrared small target semantics and the complex environment, the high-level semantics distribution is prone to overfitting the classifier. We use subpixel convolution and feature enhancement to express it as a possibility. It can effectively avoid the above problems and effectively utilize high-level semantics

The overall module $F(x, y) \in R^{H \times W \times C}$ is shown in Figure 2, and the formula is as follows,

$$F(x, y) = y + (x \cdot \sigma(C1D(\tau(\sigma(x))))) \cdot \sigma(\beta(PWConv2(\delta(\beta(PWConv1(\gamma(y))))))), \quad (1)$$

where $\sigma$, $\tau$, C1D, $\beta$, PWConv, $\delta$ and $\gamma$ are sigmoid function, global maximum pooling, 1D convolution, BN, pointwise convolution, ReLU function and global average pooling, respectively. $F(x, y) \in R^{H \times W \times C}$ has the same size as the input $X, Y \in R^{H \times W \times C}$, where $x$ is the feature map output after high-level convolution, and y is the feature map output by low-level convolution, described with the concept of modularity and can be expressed as follows:

$$Z(X, Y) = Y + P(X) \otimes L(Y). \quad (2)$$

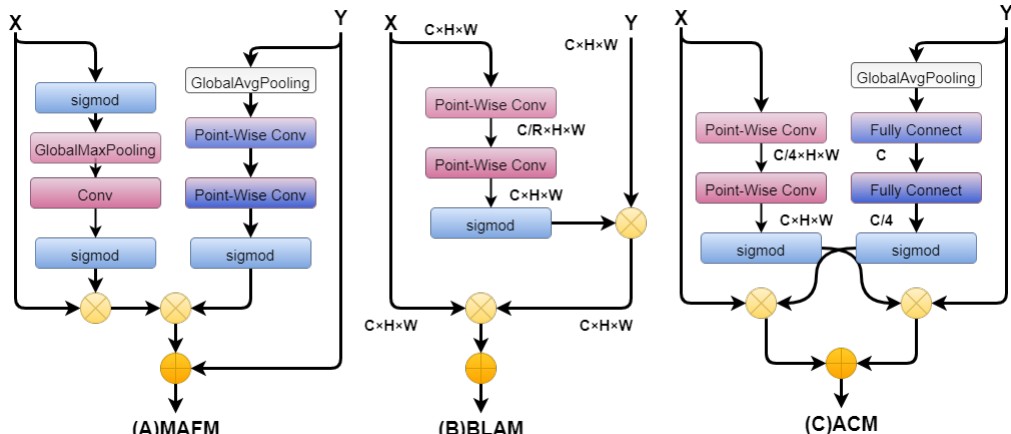

**Figure 2.** Illustration of different fusion methods. Although the three fusion structures look very similar, they are fundamentally different in terms of design concepts.

This module fuses low-level detail features of the image with deep semantic features. Because of their lack of intrinsic features, small infrared targets will be submerged in the deep layers of the network. Therefore, this study selects the underlying feature map $Y$ as the fusion benchmark and then uses the weak target channel attention mechanism to perform semantic enhancement on the deep feature map $X$ by using global maximum pooling. The obtained result $P(X)$ is regarded as the probability of the small target appearing in the receptive field, which is used as the weight of the underlying feature $L(Y)$ to guide the network to dynamically select detailed features from the underlying layer. After adding the local contrast to the original bottom feature map and deep feature map, MAFM enhances the bottom detail feature by using PA, and the deep semantic feature using WCA and the enhanced semantic feature guides the network to dynamically select detail features. Finally, a complete fused feature map $Z'$ is obtained, and $\psi$ is the subpixel convolution:

$$Z' = MLC(Y) + P(\psi(MLC(X))) \otimes L(MLC(Y)). \quad (3)$$

### 3.2.1. Weak Target Channel Attention Mechanism

With the deepening of network layers, neural networks can better understand the meaning of scenes and extract better semantic features, which helps the network distinguish

background clutter from the target. However, with the deepening of the network, the possibility of losing the target information increases. To solve this problem, we propose WCA to enhance the deep semantic information and guide the network to dynamically select the underlying details. Its process is shown in Figure 3.

We denote the output of the convolution block in the previous layer as $\chi \in R^{C \times H \times W}$, where C, H, and W are the number of channels of the feature map, height, and width. Then, the weight $p(\chi)$ of the WCA channel can be obtained by using the following formula:

$$p(\chi) = \sigma(C1D(\tau(\sigma(\chi)))). \tag{4}$$

$\tau$ is the global maximum pooling, and $\tau(\chi) = max(\chi_{i,j,k}), i \in H, j \in W, k \in C$. $\sigma$ is the sigmoid function, and C1D is the 1D convolution.

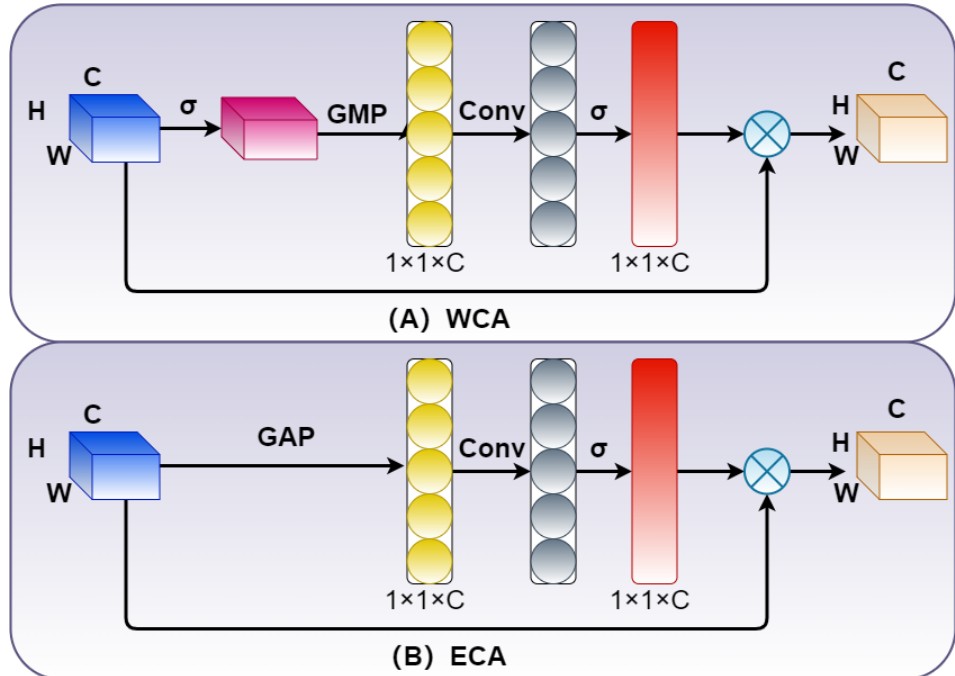

**Figure 3.** Difference between WCA and ECA. The WCA proposed by us is improved based on ECA. ECA can be better applied to large targets, but additional improvements are needed for small targets. It is mainly reflected in the addition of the sigmoid function and the global max pooling function.

The existing complex attention mechanisms perform poorly in infrared small-target detection, and lightweight attention mechanisms, such as ECA [21] cannot effectively form attention for small targets lacking in features. WCA combines the concept of lightweight attention mechanism, uses $\sigma(\chi)$ to normalize feature maps, and uses global maximum pooling to highlight deep semantic information to achieve effective attention for small infrared targets. At the beginning of the WAC, because the discrimination based on local contrast is based on the relative difference rather than the absolute value between pixels, $\sigma(\chi)$ is used to normalize the feature map in this study. Moreover, it is not appropriate to use global average pooling $\gamma(\chi) = \frac{1}{W \times H} \sum_{i=1,j=1}^{W,H} \chi_{i,j}$ because of the few intrinsic features and weak semantic information of infrared small targets. However, under the concept of treating small targets as local sparse matrices, small targets should have singular values in the local image; therefore, we use the global maximum pooling to enhance the small infrared target. Moreover, inspired by [21],we note that avoiding a reduction in the number of channels to preserve information is more important than considering channel compression without a linear relationship; thus, the number of channels is not compressed in the weak target channel attention mechanism designed in this study.

3.2.2. Pixel Attention Mechanism

In the task of infrared small-target detection, the targets are small areas in the entire image, with missing shape and texture features. Therefore, we select pointwise convolution (PWConv) to aggregate local context, which interacts with the spatial information of the local channel so that the network pays more attention to the information features with local high contrast, to highlight the small infrared targets. As shown in Figure 4, the local feature context $L$ of a pixel can be represented by pixel attention as follows,

$$L(Y) = \sigma(\beta(PWConv2(\delta(\beta(PWConv1(\gamma(Y))))))), \tag{5}$$

where $\gamma$, PWConv, $\beta$, $\delta$, and $\sigma$, represent global average pooling, pointwise convolution, BN, ReLU function, and sigmoid function, respectively. The PWConv1 and PWConv2 constitute a bottleneck structure. The pixel information of local channels is aggregated by pointwise convolution, and small infrared targets are strengthened at the pixel level, effectively solving the problem of small-target pixels and missing shape and texture features. In particular, the attention weight map $L(Y)$ has the same shape as the deep reinforcement feature map $P(X)$, so it can be fused with $P(X)$ by using the method of element multiplication. The deep semantic information can guide the network to dynamically select the bottom detail information to further enhance the bottom small target. We have

$$Y' = P(X) \otimes L(Y). \tag{6}$$

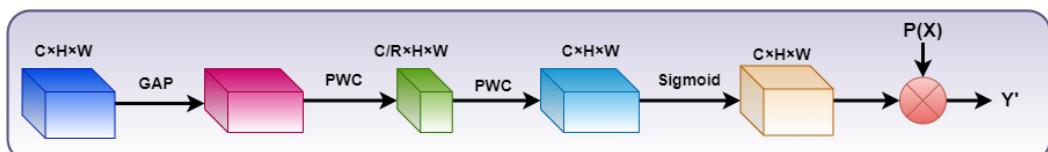

**Figure 4.** Illustration of pixel attention mechanism. The PA module is mainly used to reweight the feature map.

*3.3. Subpixel Convolution Upsampling*

After the feature maps x1, x2, and x3 of different sizes are generated through the ResNet backbone network, upsampling is required to adjust the size of the feature map for feature fusion through MAFM. The sampling methods commonly used with bilinear interpolation, such as the nearest neighbor and mean value interpolation method, have poor ability to retain image detail information because the target details information demand is higher in infrared small-target detection tasks. Therefore, inspired by [41], introducing the concept of superresolution, the subpixel convolution technique was used to obtain the sampling. This method takes a low-resolution feature map as input and obtains a high-resolution feature map through multichannel pixel reorganization and convolution. It combines individual pixels on the low-resolution multichannel feature map into a high-resolution unit of the feature map; each pixel on the low-resolution feature map acts as a subpixel on the high-resolution feature map. It can concatenate $r^2$-channel low-resolution feature maps into single-channel high-resolution feature maps. $r$ represents the multiple of upsampling of the image. This approach is shown in Figure 5.

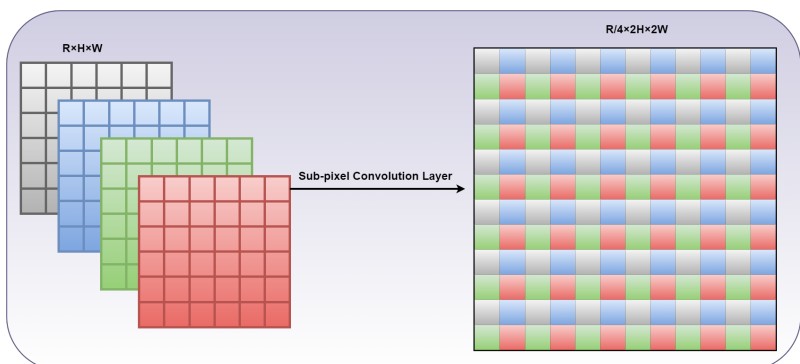

**Figure 5.** Subpixel convolution upsampling. Subpixel upsampling is mainly used to reconstruct high-quality feature maps in the decoder stage.

*3.4. Normalized Loss Function*

Although ALCNet and other methods that combine model- and data-driven detection of small infrared targets integrate the physical mechanism of the model-driven methods and the feature learning ability of the neural networks, they fail to fully consider the influence of adding a local contrastive feature refinement layer on other structures of the network and directly adopt structures, such as general loss functions, resulting in limited functions of these structures. Moreover, in the task of infrared small-target detection using local contrast, small target, and background recognition generally depend on a relative value relationship, but the traditional loss function uses the loss caused by the difference of absolute value, which is different from the contrast-based method. The traditional loss for image output values $p_{i,j}$ is defined as follows:

$$p_{i,j} = \sigma(MAF(f, \theta)). \tag{7}$$

The distribution of $p_{i,j}$ is the same as the sigmoid activation function, but its response to the loss caused by the target and background is different. Therefore, we proposed a new measurement method called normalized loss function. First, the output $p_{i,j} \in R_{H \times W}$ of the last layer of the network is probabilized as follows:

$$p_{i,j} = \frac{e^{\sigma(MAF(f,\theta))} - min(e^{\sigma(MAF(f,\theta))})}{max(e^{\sigma(MAF(f,\theta))}) - min(e^{\sigma(MAF(f,\theta))})}. \tag{8}$$

$MAF(f, \theta)$ is normalized by using maximum and minimum probability. Maximum and minimum normalization reflects relative differences and is not affected by the absolute values; it can make the background tend to 0 and the small infrared target tend to 1, to calculate the loss more accurately.

After probabilizing $p_{i,j}$ together with solving the imbalance problem between the target and background, we improve the Soft-IoU loss function into a normalized loss function, which is defined as follows,

$$\iota_{soft-iou}(p, y) = \frac{\sum_{i,j} p_{i,j} \cdot y_{i,j}}{\sum_{i,j} p_{i,j} + y_{i,j} - \sum_{i,j} p_{i,j} \cdot y_{i,j}}, \tag{9}$$

where $y_{i,j} \in R_{H \times W}$ is the label image, whose value is 0 or 1, $p_{i,j} \in R_{H \times W}$ is the probabilistic output. By transforming the prediction into probability, the relative Soft-IoU loss function that is formed can be accurately adapted to the detection method based on local contrast and has certain universality. Moreover, it can accurately reflect the position and shape of the segmented small targets.

## 4. Experiments and Results

In this section, we evaluate the effectiveness of MDFENet through experiments. First, we describe the experimental setup, including the datasets, comparison network, evaluation metrics, and implementation details. Then, we visually and numerically compare MDFENet with other model- and data-driven methods to further evaluate the performance of MDFENet. Finally, the ablation of each module of the network is studied to verify its effectiveness.

### 4.1. Experimental Settings

- Dataset: We adopted an open-source infrared small target dataset SIRST and a dataset IDTAT from the National University of Defense Technology. SIRST contains raw images and pixel-level labeled images, including 427 different images and 480 scene instances from hundreds of real videos. We divided SIRST into three sets: training set, validation set, and testing set allocated as 50%, 20%, and 30%, of the total data, respectively. The SIRST dataset is shown in Figure 6, where it can be seen from (c)(g)(h)(i)(j) that the overall image is dark, and the small target is in a complex background. The background is fuzzy, and there are clouds, Rayleigh noise, and other clutter interference. From IDTAT, we selected folders 6–12, 15, 17, and 21 with relatively complex backgrounds and 5993 pictures. The IDTAT dataset is shown in Figure 7, and its scenes are diverse and very complex. The dataset was divided into training and testing sets at a ratio of 8:2. To make the dataset applicable to the evaluation index of the network, we improved the dataset and marked the selected images at the pixel level. The experiment shows that the improved IDTAT can be effectively applied to the network.

- Comparison Network: To prove the effectiveness of MDFENet, we compared the proposed method with other model- and data-driven methods. From the data-driven methods, we selected FPN [31], the attention local contrast network [19], and the attention-guided pyramid environment network [17]. We select these data-driven methods for comparison because our network was inspired by the above networks and proposed our new models and ideas on this basis. Therefore, we select the above networks as the baseline to make a fair comparison and prove the superiority of our proposed models and ideas. These methods have the same optimizer and other hyperparameters as the proposed method. In the traditional method, two classical methods and two new methods are selected, which are MPCM algorithm [10], IPI algorithm [14], PSTNN algorithm [18], and FKRW algorithm [26]. MPCM and IPI are classical algorithms in infrared small-target detection and have a relatively wide influence. PSTNN and FKRW are advanced model-driven methods in recent years, and their comparison can better prove the superiority of our method.

- Evaluation Metrics: This algorithm is a segmentation-based target-detection algorithm, and the result is a segmented binary image. Therefore, we do not consider traditional infrared small-target detection evaluation indicators, such as background suppression factor and signal-to-noise ratio, to be applicable. Instead, we used FLOPs, Params, IoU, and the normalized intersection over union (nIoU) to objectively evaluate the performance of the proposed network. FLOPs and Params are the key metrics by which to evaluate network speed and lightweightness. IoU is an important metric in pixel segmentation tasks to evaluate the shape detection ability of an algorithm. nIoU is an infrared small-target detection evaluation metric proposed by [19] , which can better balance the metrics between model- and data-driven methods, which is defined as

$$nIoU = \frac{1}{N} \sum_{i}^{N} \frac{TP[i]}{T[i] + P[i] - TP[i]},$$

(10)

where T, P, and TP denote true, positive and true positive, respectively.

- Runtime and Implementation Details: Table 2 shows the computational complexity and running time of deep-learning methods. ResNetFPN, ALCNet, AGPCNet and

our proposed MDFENet methods based on deep learning are trained on a laptop computer with Nvidia GeForce RTX 2080 Super GPU and 8 G GPU memory, and the code is written in Python 3.8 language using the Pychram 2021 IDE and PyTorch 1.8 framework. When we implemented these methods, we used the ADAM optimizer, where the weight decay coefficients were set to 0.0001. The initial learning rate was 0.001, and the decay strategy of poly was used. The batch size was set to 8. The loss function for each method was normalized Soft-IoU. SIRST dataset was trained for 300 epochs, and the IDTAT dataset was trained for 40 epochs. The model-driven methods, MPCM, IPI, PSTNN, and FKRW were run on a laptop computer with 2.20 Ghz CPU Intel(R) Core(TM) i7-10870H and 16 G main memory. The code was implemented in MATLAB R2021b. The specific experimental parameter settings of each method during implementation are shown in Table 3.

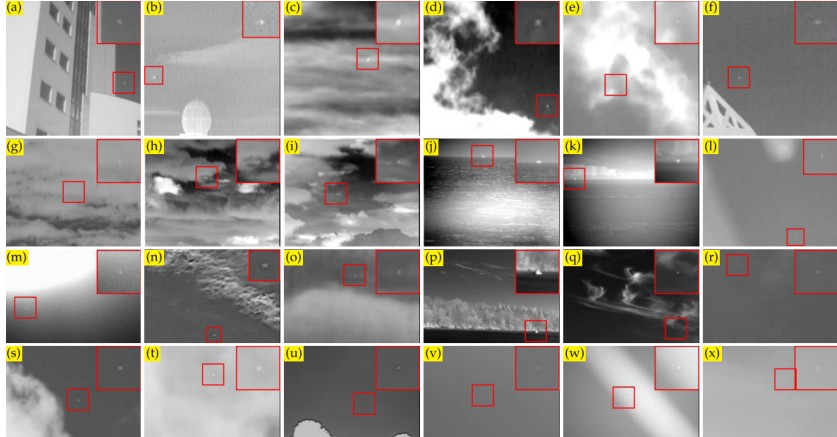

**Figure 6.** Partial presentation of the SIRST dataset. The images (**a–x**) come from several different scenes and the number of pixels occupied by the target is very small. The small red box in the image is the area where the target is located, and it will be enlarged and displayed in the big red box in the upper right corner.

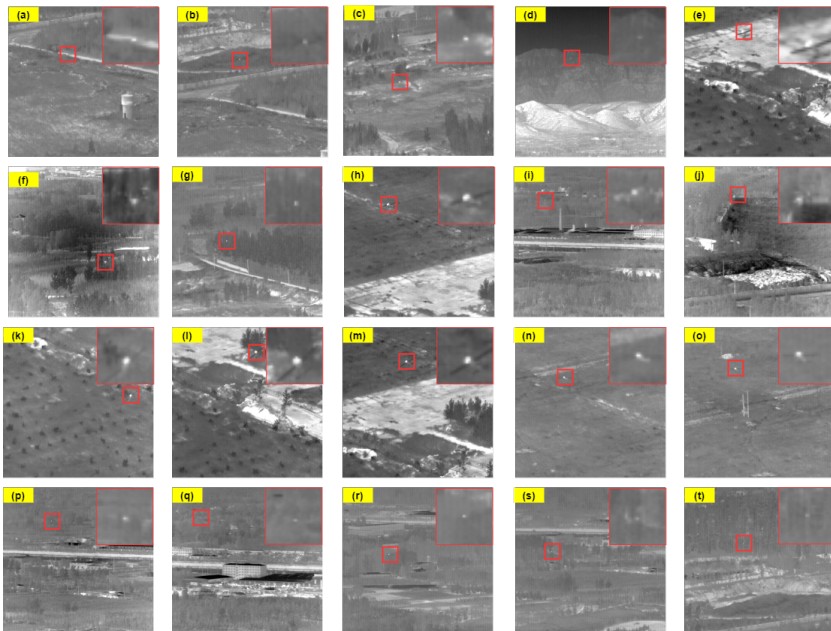

**Figure 7.** Partial presentation of the IDTAT dataset. The images (**a–t**) come from several different scenes and the number of pixels occupied by the target is very small. The small red box in the image is the area where the target is located, and it will be enlarged and displayed in the big red box in the upper right corner.

**Table 2.** Computational complexity and running time of deep learning algorithms.

|  | **ResNetFPN** | **ALCNet** | **AGPCNet** | **MDFENet** |
|---|---|---|---|---|
| FLOPs | 16.105 G | 4.336 G | 50.602 G | 4.348 G |
| Params | 0.546 M | 0.372 M | 12.360 M | 0.383 M |
| Times(s) | 0.0261 | 0.0223 | 0.0462 | 0.0231 |

**Table 3.** Hyperparameter settings of model-driven methods.

| Method | Hyperparameter Settings |
|---|---|
| MPCM | N = 1, 3, . . . , 9 |
| IPI | PS: $50 \times 50$, stride: 10, $\lambda = \frac{L}{min(m,n)^{1/2}}$, $L = 4.5$, $tf : k = 10$, $\xi = 10^{-7}$ |
| PSTNN | patchSize = 40, slideStep = 40, lambdaL = 0.7 |
| FKRW | K = 4, p = 6, $\beta = 200$, window size = $11 \times 11$ |

*4.2. Comparison to State-of-the-Art Approaches*

To demonstrate the superiority of our method, we compared MDFENet with other state-of-the-art data-driven based methods and traditional model-driven methods in both quantitative and qualitative aspects, and the results are shown in Tables 2 and 4, Figures 8–14.

Quantitative results: Tables 2 and 4 present the results of the different methods. Clearly, our proposed MDFENet network was at the forefront of lightweight methods and achieved the best detection results; the improvement effect was noticeable. From the quantitative results, the performance of data-driven algorithms was better than those of model-driven algorithms, and the performance of the model-driven algorithms was not ideal. The reason lies in two aspects.

1. The model-driven algorithms have strong assumptions about the environment whereas the environments of SIRST and IDTAT datasets are complex and changeable; therefore, the assumptions of the model-driven algorithms are not fully satisfied, leading to poor performance.

2. The classical infrared small-target detection method does not completely segment the target but detects the position of the target. The target detected by these methods is often incomplete; therefore, the performance is poor.

Model-driven algorithms face a more serious problem—the generation of false alarms, which causes many problems when directly applied to the field of early warning. In contrast, because data-driven methods are not bound by model assumptions, the results of the network depend on the learned features, which can achieve accurate segmentation and reduce false alarms. Moreover, data-driven methods can perform feature fusion. The existing networks have feature fusion modules, but the method of feature fusion is different. The multimechanism attention collaborative fusion module designed in this study for the characteristics of small infrared targets with few intrinsic features and small-target proportions was effective. Experimental data show that our proposed network outperformed other advanced deep learning methods in suppressing the background, accurately detecting targets, and segmenting targets.

Qualitative results: As shown in Figures 8–13, we present the detection results of images with very small dim targets and very complex backgrounds in the dataset and compare the detection results of the eight methods. The target area is enlarged for better display. Red and blue quadrangles represent correct targets and false alarms, respectively.

As can be seen from the results, our MDFENet achieved accurate target localization output and shape segmentation. Model-driven methods are sensitive to noise and detected more false alarm areas. Other state-of-the-art data-driven-based methods suffered from missed detections and false-alarm regions under the condition that small targets were extremely weak, and our MDFENet was more robust to these complex scenes. Furthermore,

from the perspective of shape segmentation, our MDFENet produced more accurate shape segmentation and achieved better performance than other advanced data-driven methods.

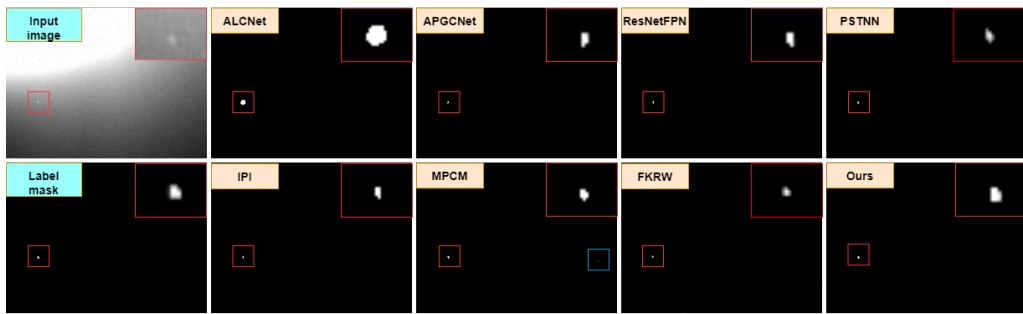

**Figure 8.** Detection results of infrared small targets in Scene 1 using different detection methods. The target area is enlarged for better display. Red and blue squares represent correct targets and false alarms, respectively.

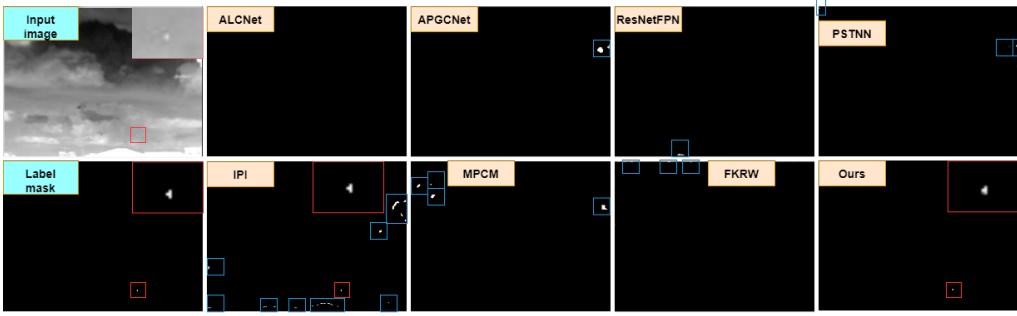

**Figure 9.** Detection results of infrared small targets in Scene 2 using different detection methods.

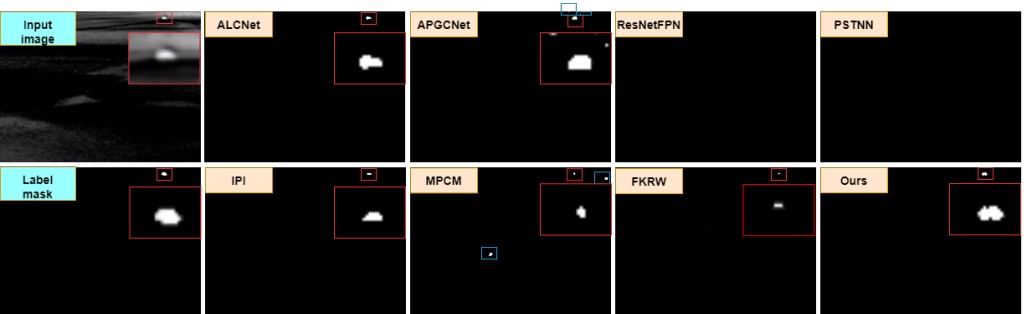

**Figure 10.** Detection results of infrared small targets in Scene 3 using different detection methods.

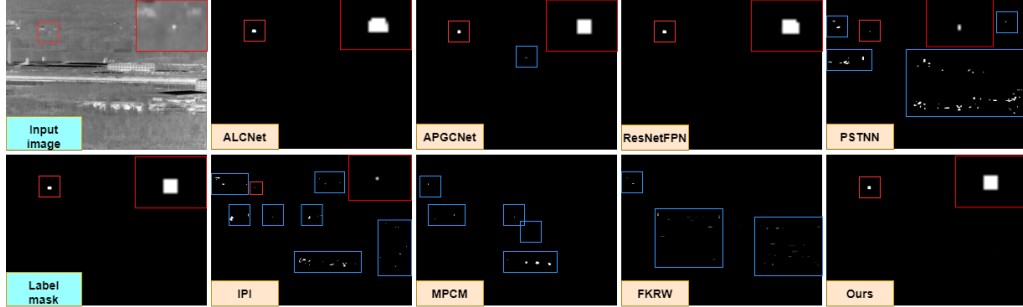

**Figure 11.** Detection results of infrared small targets in Scene 4 using different detection methods.

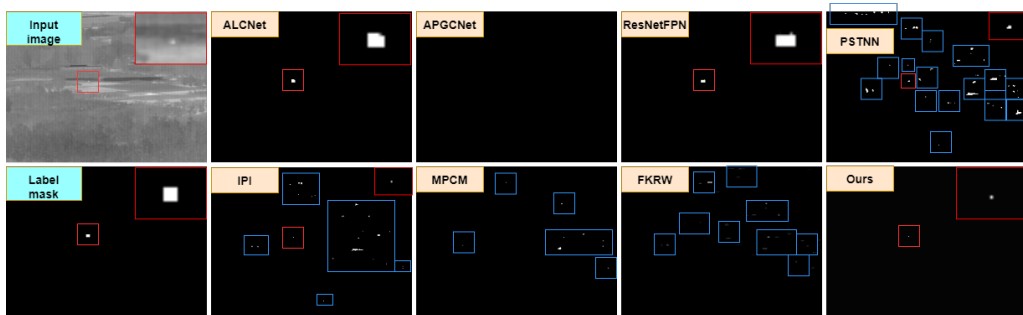

**Figure 12.** Detection results of infrared small targets in Scene 5 using different detection methods.

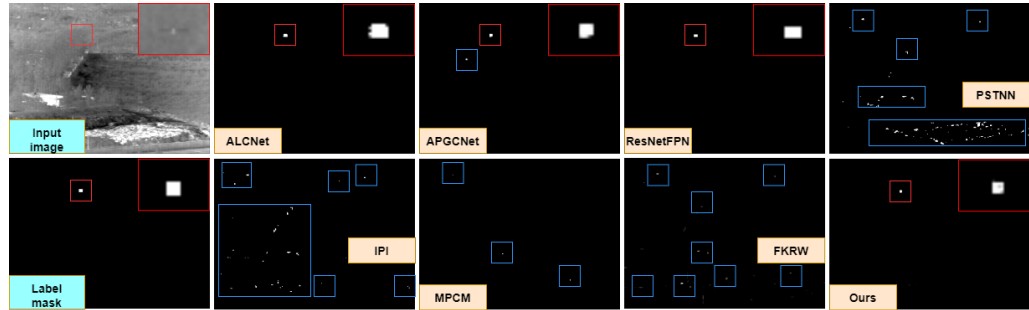

**Figure 13.** Detection results of infrared small targets in Scene 6 using different detection methods.

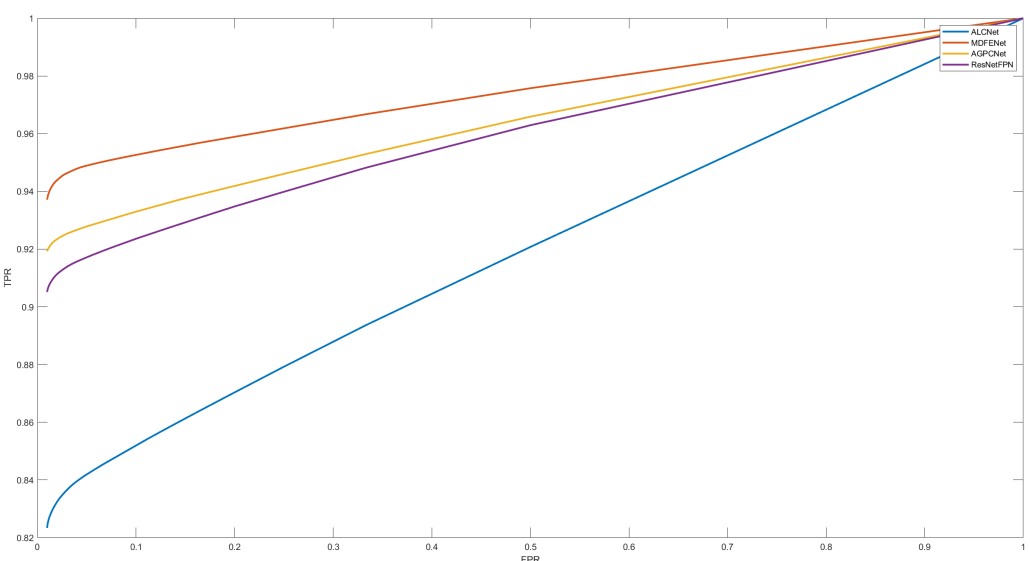

**Figure 14.** ROC curve on dataset.

**Table 4.** Performance comparison results with other model- and data-driven methods.

| Dataset | Metric | PSTNN | FKRW | MPCM | IPI | ResNetFPN | ALCNet | AGPCNet | MDFENet |
|---|---|---|---|---|---|---|---|---|---|
| SIRST | IoU | 0.596 | 0.205 | 0.493 | 0.720 | 0.757 | 0.762 | 0.799 | **0.809** |
| | nIoU | 0.577 | 0.253 | 0.351 | 0.650 | 0.751 | 0.761 | 0.795 | **0.808** |
| IDTAT | IoU | 0.019 | 0.019 | 0.013 | 0.017 | 0.761 | 0.803 | 0.779 | **0.825** |
| | nIoU | 0.010 | 0.008 | 0.009 | 0.023 | 0.759 | 0.801 | 0.778 | **0.823** |

The best performance values are shown in bold. The following tables are the same.

### 4.3. Ablation Study

To better show the performance of each part of the MDFENet network, we carried out ablation experiments by removing or replacing several specific parts of MDFENet.

1. Influence of multimechanism attention collaborative feature fusion module. Tables 5 and 6 present the different feature fusion methods and strategies we adopted. Figure 15 show the structure diagrams of each fusion method. We selected the bottom-up attentional module (BLAM) and asymmetric contextual modulation (ACM) and multimechanism top-down attention module (MTAM) and multimechanism reverse attention module (MRAM) integrating channel attention mechanism [21]. We replaced the MAFM with the above modules to verify the effectiveness of the MAFM. As shown in Tables 5–7, SIRST and IDTAT were selected as the datasets. Our proposed MAFM has the optimal effect.

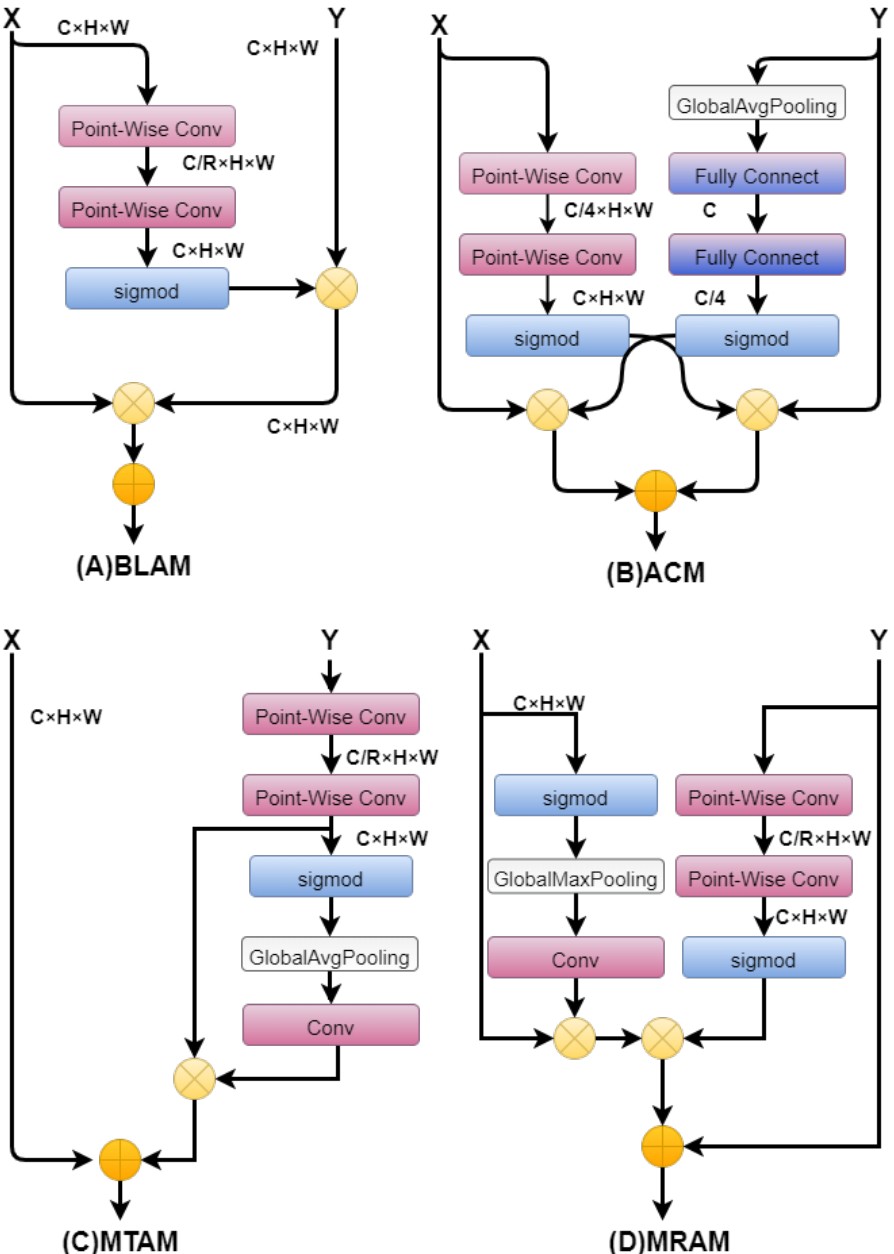

**Figure 15.** Architecture of various fusion modules. (**A**) Bottom-up attentional module (BLAM). (**B**) Asymmetric contextual modulation (ACM). (**C**) Multimechanism top-down attention modules(MTAM). (**D**) Multimechanism reverse attention module (MRAM).

**Table 5.** Performance comparison of different fusion modules on the SIRST dataset.

| Manner | IoU | | | | nIoU | | | |
|---|---|---|---|---|---|---|---|---|
| | b = 1 | b = 2 | b = 3 | b = 4 | b = 1 | b = 2 | b = 3 | b = 4 |
| BLAM | 0.745 | 0.769 | 0.781 | 0.799 | 0.758 | 0.772 | 0.779 | 0.785 |
| ACM | 0.754 | 0.763 | 0.784 | 0.796 | 0.749 | 0.756 | 0.780 | 0.795 |
| MTAM | 0.748 | 0.773 | 0.778 | 0.788 | 0.749 | 0.771 | 0.777 | 0.781 |
| MRAM | 0.735 | 0.760 | 0.771 | 0.796 | 0.734 | 0.764 | 0.769 | 0.793 |
| MAFM | 0.756 | 0.787 | 0.802 | **0.809** | 0.757 | 0.782 | 0.799 | **0.808** |

**Table 6.** Performance comparison of different fusion modules on the IDTAT dataset.

| Manner | IoU | | | | nIoU | | | |
|---|---|---|---|---|---|---|---|---|
| | b = 1 | b = 2 | b = 3 | b = 4 | b = 1 | b = 2 | b = 3 | b = 4 |
| BLAM | 0.806 | 0.810 | 0.805 | 0.812 | 0.805 | 0.808 | 0.804 | 0.812 |
| ACM | 0.803 | 0.814 | 0.815 | 0.819 | 0.802 | 0.814 | 0.815 | 0.818 |
| MTAM | 0.811 | 0.815 | 0.818 | 0.820 | 0.809 | 0.814 | 0.818 | 0.819 |
| MRAM | 0.808 | 0.810 | 0.819 | 0.813 | 0.807 | 0.810 | 0.818 | 0.819 |
| MAFM | 0.810 | **0.826** | 0.823 | 0.825 | 0.808 | **0.824** | 0.821 | 0.823 |

**Table 7.** Ablation study on the whole network.

| Backbone | BLAM | MAFM | Sub-Pixel | NL | SIRST | | IDTAT | |
|---|---|---|---|---|---|---|---|---|
| | | | | | IoU | nIoU | IoU | nIoU |
| ResNet-20 | ✓ | | | | 0.762 | 0.761 | 0.803 | 0.801 |
| ResNet-20 | | ✓ | | | 0.766 | 0.771 | 0.806 | 0.805 |
| ResNet-20 | | ✓ | ✓ | | 0.780 | 0.775 | 0.817 | 0.816 |
| ResNet-20 | | ✓ | ✓ | ✓ | **0.809** | **0.808** | **0.825** | **0.823** |

2. Influence of sub-pixel convolutional upsampling. A high-resolution prediction map can effectively improve the effect of infrared small-target detection. In this study, we used subpixel convolution in a superresolution reconstruction task to improve the resolution of the prediction map. We studied and compared subpixel convolution and linear interpolation in this study. As shown in Table 7, the subpixel convolution network performance is outstanding. This is because of the traditional linear interpolation in spatial information only and in the neural network; correlation and interaction between each channel cannot be ignored, and the subpixel convolution can be a relatively good application of these relations through channel expansion. It is then reconstituted into the image space and the information between channels is effectively used to achieve better results than traditional sampling.

3. Influence of normalized loss function. In the traditional IoU and nIoU calculation, the loss calculation of the network output results directly after the sigmoid and the labeled image cannot accurately reflect the error. The normalized probability function greatly reduces the redundant information in the image, suppresses the background, enhances the target, and enables the network to learn more accurate information. The results are presented in Table 7. The performance of the network was greatly improved after the normalized loss was adopted.

## 5. Conclusions

In this study, a new detection method, MDFENet, was developed. The network was based on combining data- and model-driven methods by using lightweight structure design. MAFM was constructed to enhance the fusion of multiscale features and make full use of infrared small target detail features and semantic features. This study draws on the concept of superresolution and adopts a subpixel convolution method for feature upsampling to

improve the resolution of the feature map and further improve the detection accuracy. Finally, the normalized loss function was designed to normalize the network output by maximum and minimum probability so that the network could make use of the relative difference between the target and the background during detection. This enabled the model to calculate the loss more accurately and improved the network performance. Experimental results on two datasets—the SIRST dataset, and the improved IDTAT dataset—showed that each module of the MDFENet network was effective. The network outperformed the traditional model- and advanced data-driven methods, suggesting that the fusion of the features of different layers after the attention weighting can more effectively improve the performance of the neural network to detect small infrared targets. Additionally, methods that combine model- and data-driven detection should design a specific normalized loss function, which may have better detection results.

However, our method is essentially a single-frame target detection based on segmentation. Compared with the multiple-frames-based sequential detection method [42–45], it cannot maximize the information between the target and the background and the accumulation of motion energy. In addition, although our approach achieves an excellent balance between lightweightness and detection performance, its efficiency and stability in actual deployment have not been proven. In future work, we will further study how to apply the proposed feature-fusion enhancement strategy and standardized loss function to improve the feature fusion efficiency and detection capability of multiframe sequence detection methods and focus on the application of high-efficiency infrared small-target detection network in actual deployment.

**Author Contributions:** Conceptualization, Y.Z. (Yi Zhang) and B.N.; methodology, Y.Z. (Yi Zhang) and B.N.; software, Y.Z. (Yi Zhang), B.N., Y.Z. (Yu Zhang) and F.L.; supervision, Y.Z. (Yan Zhang) and F.L.; validation, Y.Z. (Yi Zhang), B.N. and Y.Z. (Yan Zhang); formal analysis, Y.Z. (Yan Zhang), Y.Z. (Yu Zhang) and F.L.; data curation, Y.Z. (Yi Zhang) and B.N.; writing—original draft preparation, Y.Z. (Yi Zhang) and B.N.; writing—review and editing, Y.Z. (Yi Zhang), Y.Z. (Yu Zhang) and Y.Z. (Yan Zhang); project administration, Y.Z. (Yi Zhang). All authors have read and agreed to the published version of the manuscript.

**Funding:** This work was supported by The National Natural Science Foundation of China (No. 62075239).

**Data Availability Statement:** The data used for training and test set SIRST are available at: https://github.com/YimianDai/sirst, accessed on 22 August 2022.

**Conflicts of Interest:** The authors declare no conflict of interest.

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
