# Peer review of "Lightweight Multimechanism Deep Feature Enhancement Network for Infrared Small-Target Detection"

_remotesensing, doi:10.3390/rs14246278_

Round 1

Reviewer 1 Report (Previous Reviewer 3)

This paper is well written. There are some flaws in this paper.

1.     In the Introduction, the author mentioned that ResNet20 is selected as the backbone network. Why ResNet20? The reason should be given.

2.     Why these comparison algorithms are selected should also be explained briefly.

3.     The computer information and software used in the experiment should be given to facilitate the reader to restore the experiment.

Author Response

Reviewer 2 Report (Previous Reviewer 2)

Congratulations on all improvements done to your research. 

Author Response

Dear reviewer:

Thank you very much for your kind work and consideration on the improvement of our paper.

Reviewer 3 Report (Previous Reviewer 1)

The authors have addressed all my comments. I recommend this paper for publication.

Author Response

Dear reviewer:

Thank you very much for your kind work and consideration on the improvement of our paper.

This manuscript is a resubmission of an earlier submission. The following is a list of the peer review reports and author responses from that submission.

Round 1

Reviewer 1 Report

Specific to the problem of infrared small target detection in complex background, the authors proposed a multi-mechanism deep feature enhancement network model. However, there is still a room for improvement in the paper organization and presentation. More specifically, there are several issues that need to be addressed before a possible publication.

1.       It is better not to use "target" or "object" alternately. Please check the whole article.

2.       There may be some grammatical or spelling errors in the following statements:

Page 2 Line 67: “...tends to leads to... ”,“ leads” should be “ lead”.

Page 2 Line 70: “... data analysis and can be ... ”, "and" should be deleted.

3.       The baseline methods used in this paper, such as IPI and MPCM, are old, and only five baseline methods are insufficient. Therefore, the author is suggested to compare and add the latest infrared small target detection methods. Please refer to but not limited to the following literature:

[1]      L Zhang, Z Peng. Infrared Small Target Detection Based on Partial Sum of the Tensor Nuclear Norm[J], Remote Sens., 2019,11(4), 382.

[2]      Y. Qin, L. Bruzzone, C. Gao, and B. Li, “Infrared small target detection based on facet kernel and random walker,” IEEE Trans. Geosci. Remote Sens., 2019.

4.       Personally, in 4.1. Experimental Settings, the dataset source has been given, and the presentation of the dataset in Figure 6 and 7 is optional. However, Figure 8-10 only shows the visual detection results of three relatively simple scenes, and readers may pay more attention to the detection effect in complex scenes. Please consider adding more visual detection results in complex backgrounds.

5.       In addition, false alarms in the detection results have been marked with blue rectangles. Is 3D visualization of detection results in Figure 11-13 necessary?

6.       It is very meaningful to add the ROC or PR curves. For details, please refer to the following literature.

[1]      Dai, Y., et al., Attentional Local Contrast Networks for Infrared Small Target Detection. IEEE Transactions on Geoscience and Remote Sensing, 2021. 59(11): p. 9813-9824.

7.       The author is suggested to look forward to the limitations of the proposed method in the conclusion. For example, multiple frames-based the sequential detection method are very popular at present, and more target and background information can be considered. Please refer to the following latest literature:

[1]      A spatial-temporal feature-based detection framework for infrared dim small target. IEEE Trans. Geosci. Remote Sens. 3, 117–131 (2021)

[2]      D. Pang, T. Shan, W. Li, P. Ma, R. Tao, and Y. Ma, “Facet derivativebased multidirectional edge awareness and spatial-temporal tensor model for infrared small target detection,” IEEE Trans. Geosci. Remote Sens., pp. 1–15, 2021.

[3]      H. Liu, L. Zhang, and H. Huang, “Small target detection in infrared videos based on spatio-temporal tensor model,” IEEE Trans. Geosci. Remote Sens., vol. PP, no. 99, pp. 1–12, 2020.

[4]      D. Pang, T. Shan, P. Ma, W. Li, S. Liu, and R. Tao, “A novel spatiotemporal saliency method for low-altitude slow small infrared target detection,” IEEE Geosci. Remote Sens. Lett., pp. 1–5, 2021.

Reviewer 2 Report

Taking into account the possibility that this manuscript does get published, I think it is appropriate to include some brief comments. 

- The manuscript seems somewhat with grammatical/syntax and typographical problems. I leave it to the authors to resolve these copyediting problems by actually thoroughly reading the manuscript. Problems of this sort should definitely not appear in print. 

Congratulations on your ongoing/concluded research. 

Reviewer 3 Report

This paper has proposed Lightweight Multi-mechanism Deep Feature Enhancement

Network for Infrared Small Target Detection. I think it was well written. There are some flaws in this paper.

1.     In terms of article structure, sections 2 and 3 should be combined into one section. Or section 2 can be abbreviated and placed in Section 1. Now section 2 and section 3 are very relevant, so there is no need to separate them. What's more, section 2 is more like the introduction.

2.     Section 3.1. Network Structure should be described in more detail so that readers can better understand the algorithm process.

3.     The computational complexity of the proposed algorithm needs to be given, or the running time should be given.

4.     This article lacks the introduction of the computer equipment and software platform of the experiment, which is not conducive to the readers to restore the experiment.